# Risk Perception, Risk Preference, and Timing of Food Sales: New Insights into Farmers’ Negativity in China

**DOI:** 10.3390/foods13142243

**Published:** 2024-07-16

**Authors:** Tan Tian, Xia Zhao

**Affiliations:** Institute of Food and Strategic Reserves, Nanjing University of Finance and Economics, #17A, No.3, Wenyuan Road, Nanjing 210003, China; ttryu@126.com

**Keywords:** food price risk, risk perception, risk preference, farmers’ food sales

## Abstract

Chinese farmers, especially small ones, always sell their food at low prices during harvest season rather than storing it for a better price. Based on a theoretical framework of expected utility, this paper examines the mechanism by which risk perception affects farmers’ timing choices of food sales and the role played by risk preference, utilizing data from the 2019 China Family Database and the China Household Finance Survey of farmers in six provinces of the main wheat-producing regions. This study shows that farmers with a high risk perception are more likely to choose current sales compared with intertemporal sales. The channel and mechanism analysis finds that increased risk perception leads to risky returns from intertemporal sales lower than certain returns from current sales in utility comparisons. It is further found that risk preference has a substitution effect on risk perception in farmers’ intertemporal food sales.

## 1. Introduction

Food prices in developing countries show a cyclical pattern of seasonal fluctuations [1,2]. In a typical normal year, food prices are lowest during the harvest season and then gradually rise and peak in the lean season before the next harvest. With new foods coming to the market, prices decline and enter the next volatility cycle [3]. The price fluctuation pattern is mainly related to the food production cycle and the seasonality of supply, which objectively creates conditions for farmers’ intertemporal food sales for arbitrage.

Even so, most Chinese farmers show a negative attitude towards food sales. They sell their hard-earned food in the shortest time and most convenient way at low prices. It does not look like they care about the benefit opportunities. In 2020, China comprehensively won the battle against poverty, and 98.99 million rural poor were lifted out of poverty by the current standard [4]. Nevertheless, many small farmers are still living just above the poverty line, and food sales have the most direct impact on their production returns. Conversely, the negativity of Chinese farmers towards food sales has been observed in both academia and reality [5].

Explanations for farmers’ negativity in food sales focus on liquidity, storage, and transaction cost constraints [6,7,8]. Storing food for opportunistic sale is only feasible if there are adequate funding sources, enough storage space and facilities, and relatively low transaction costs. Other factors that may have impacts include planting size [9], storage losses [10], and risk preference [11]. However, food sales behavior cannot be simply explained by a single factor or perspective from economic interests, it is a comprehensive psychological choice under external uncertainty and is closely related to farmers’ risk perception and risk preference [12].

We contribute a new insight that farmers’ negativity in the timing of food sales is based on the risk perception of intertemporal sales under utility comparison. On the one hand, Chinese farmers’ cultivated land is fragmented on a small scale, and the income from food sales accounts for a low proportion of the total household income [13], so farmers perceive that the utility brought by intertemporal food sales is insufficient; on the other hand, most of the farmers are averse to risk [14], and based on the fact that food prices in the previous years did not always rise over time or rose at a small rate, the aversion to negative return may induce farmers to sell food negatively. Further, consider the amplification of risk perception by constraints, such as liquidity, storage, and transaction costs, which leads farmers to perceive the utility of the risk returns from intertemporal sales to be lower than the utility of the certainty returns from current sales. The negative food sale is in fact a rational, constrained “second-best” choice for farmers.

Regarding the economic behavior of farm households, it has been widely recognized in the existing literature that risk perception and risk preference are central variables in individual risk behavior decisions [15,16,17]. Using mathematical models, Pratt [15] pioneered the idea that an individual’s risk behavior depends on risk perception and risk preference. Kahneman and Tversky [16] proposed in prospect theory that risk perception and risk preference are the two main factors influencing an individual’s decision making and that they are inseparably linked. Risk perception is the decision maker’s judgment and evaluation of the magnitude of risk. Slovic [17] defined risk perception as the process by which people rely on their intuition to make risk judgments and assessments of the unknown, whereas risk preference is the psychological response of the decision maker to risk. Although the importance of risk perception and risk preference in individual risk decision making is unanimously recognized, there is still a debate on the relationship between the two and the influence of their behavioral effects [18,19,20]. Sitkin and Weingart [19] argued that risk preference affects the relative salience of situational threats or opportunities, leading to biased risk perception. Lusk and Coble [20] considered that risk perception is more influential than risk preference in the acceptance of GM foods.

We reveal the separate roles and the mechanism between risk perception and risk preference in influencing farmers’ food sale decisions. Previous studies, including prospect theory, tend to discuss the two together in a mishmash without directly examining the causal mechanisms of the behavioral effects, and a decoupled description of the two is needed if we are to accurately analyze the effects of risk perception and risk preference on farmers’ food sale behavior. Specifically, in the context of food sales by farmers, risk perception and risk preference are the main dimensions for farmers to consider the risk of food storage and sales, but for farmers who are mostly risk averse, risk perception plays a more important role in the actual food sales process. This is consistent with the findings of Kahneman and Lovallo [18], Weber et al. [21], and Lusk and Coble [20]. With the application of new technologies, farmers first systematically use the information they have gathered to assess the magnitude of the risk of food sales, rather than considering directly based on risk preference. For example, Barrett and Dorosh [22] stated that the risk perception of price uncertainty reduces the incentives for poor farmers to store food in Madagascar. Through a randomized controlled experiment, Burke et al. [8] showed that credit constraints amplified the risk perception of Kenyan farmers, leading to “sell low and buy high” for food.

The remainder of the article is organized as follows. The second part constructs the theoretical framework and conducts the mechanism analysis; the third part presents the data sources and the choice of an econometric model; the fourth part reports and analyzes the estimation results; and the last part concludes the full article.

## 2. Theoretical Framework

### 2.1. Theoretical Framework of Risk Perception and Farmers’ Timing of Food Sales

Based on the expected utility theory, this paper draws on Cardell and Michelson’s [14] analysis to develop a theoretical model of the relationship between risk perception and farmers’ timing choices for food sales. As the basis of individual decision-making theory under uncertain conditions, although some of the shortcomings have been challenged by the “Allais Paradox” and prospect theory, the expected utility theory (EUT) can still provide a more complete explanation under assumptions of rationality, risk aversion, and long-term decisions [23], while food farmers fulfill the assumptions. First, farmers tend to be rational. Farmers systematically use the information they have collected to assess and consider the impact of implementing the behavior before making decisions to sell food, and although farmers’ decisions show irrationality, they are essentially rational choices under the constraints of external conditions. Secondly, farm households are basically risk averse [24]. Farmers’ risk preference is risk averse and is considered to be a consistent and unchanging psychological trait [14]. When faced with the complexity and variety of agricultural risks, farmers tend to show more risk-averse attitudes and shy away from agricultural production activities with higher expected returns [25]. Risk minimization tends to be a more important decision-making objective for most farmers than profit maximization. Again, farmers tend to be consistent in their long-term decision making. Most farmers have much experience in agricultural activities, and their responses to losses and profits from food sales are relatively rational, and long-term practice tends to stabilize farmers’ mindsets.

First, based on the above analysis, we assume that farmers are rational, risk averse, and price takers in input and output markets and that they are in a perfectly competitive market with identical storage, credit conditions, and zero cost. At harvest time, the price $p_{h}$ is known, but the price of lean season $p_{l}$ is unknown. Storage returns $r=\frac{p_{l}-p_{h}}{p_{h}}$; if r > 0, then it should be stored, but the randomness of lean season prices exposes farmers to price risk and, realistically, farmers face additional costs, such as credit and storage. We assume that farmers have Von Neumann–Morgenstern (VNM) utility function to calculate the deterministic equivalent rate of return for the farmer. The risk premium equation is as follows:(1)$$E\left[U\left(w\left(1+r\right)\right)\right]=U\left(w\left(1+C\right)\right)$$
where w represents the farmer’s wealth, r represents the risk–return ratio of intertemporal sales, and C is the certainty equivalent rate of the return that the farmer is willing to forgo regarding intertemporal sales. Assuming that the utility function $U(w\left(1+r\right))$ is a second-order continuous derivable, a second-order Taylor expansion on the left side of the equal sign in Equation (1) at the mean $\;w\left(1+\bar{r}\right)$ yields the following:(2)$$U\left[w\left(1+r\right)\right]\approx U\left[w\left(1+\bar{r}\right)\right]+U^{\prime }\left[w\left(r-\bar{r}\right)\right]+\frac{1}{2}U^{\prime \prime }{\left[w\left(r-\bar{r}\right)\right]}^{2}$$

In Equation (2), $U^{\prime }$ = $\frac{\partial_{U}}{\partial_{w}},U^{\prime \prime }$= $\frac{\partial_{U}^{2}}{\partial_{w}^{2}}$. Taking the expectation in Equation (2) yields the following:(3)$$E\left\{U\left[w\left(1+r\right)\right]\right\}\approx U\left[w\left(1+\bar{r}\right)\right]+U^{\prime }w*E\left(r-\bar{r}\right)+\frac{1}{2}U^{\prime \prime }w^{2}\sigma^{2}$$

In Equation (3), $\sigma^{2}$ is the risk–return variance of a farmer’s choice for intertemporal sales, representing farmers’ risk perception of intemporal sales. The greater the farmer’s perceived risk of intertemporal sales, the greater $\sigma^{2}.$ Since $E\left(r-\bar{r}\right)=0$, Equation (3) reduces to the following:(4)$$E\left\{U\left[w\left(1+r\right)\right]\right\}\approx U\left[w\left(1+\bar{r}\right)\right]+\frac{1}{2}U^{\prime \prime }w^{2}\sigma^{2}$$

Also, a first-order Taylor expansion on the right side of the equality sign in Equation (5) at $w\left(1+\bar{r}\right)$ yields the following:(5)$$U\left[w\left(1+r\right)\right]\approx U\left[w\left(1+\bar{r}\right)\right]+U^{\prime }\left[w\left(C-\bar{r}\right)\right]$$

Joining Equations (4) and (5), we obtain the following:(6)$$C\approx \bar{r}-\left(-\frac{1}{2}\frac{wU^{\prime \prime }}{U^{\prime }}\sigma^{2}\right)$$

We set the absolute risk aversion coefficient as $A=-\frac{U^{\prime \prime }}{U^{\prime }}$, which denotes the amount of wealth a farmer is willing to give up in order to hedge the risk of losing 1 unit quantity of wealth; the relative risk aversion coefficient, $R=-\frac{wU^{\prime \prime }}{U^{\prime }}$, denotes the proportion of wealth that a farmer is willing to give up in order to avoid the risk of a 1 percent loss of wealth. Logically, A varies greatly with the amount of individual wealth and does not completely portray the risk preference of farmers; the degree of aversion to the risk of proportional loss of wealth, R, is more reflective of the inherent attitude of farmers to risk, and this paper uses R to refer to the risk preference of farmers. A higher R value means that farmers are more risk averse and have a lower risk preference. Therefore, the certainty equivalent return C can be expressed as follows:(7)$$C\approx \bar{r}-\frac{1}{2}R\sigma^{2}$$

From the theoretical mechanism, an increase in R and $\sigma^{2}$ in Equation (7) implies a decrease in C, i.e., the value of intertemporal food sales diminishes as the farmer’s risk preference decreases and risk perception increases, thus reducing the likelihood that the farmer will choose intertemporal food sales. This leads to the following two research hypotheses:

**H_1_.** 
*The higher the farmer’s perception of risk, the less likely it is that the farmer will choose intertemporal or two-phase (two-phase sales of food means that farmers use part of their harvest for current sale and part for inter-period sale. Farmers tend to realize part of their harvest at harvest time to meet consumption needs, pay off debts, etc., while storing the rest for selling opportunities (Cardell and Michelson, 2022 [14]) sales compared to current sales.*


**H_2_.** 
*The lower the level of risk preference, the less likely a farmer is to choose intertemporal or two-phase sales compared to current sales.*


In the framework of expected utility theory and its variants, risk preference is only a descriptive label that technically refers to the curvature of the utility function. When farmers are mostly risk averse, which is considered to be a consistent and invariant psychological trait, then $U_{w}^{\prime }>0,U_{w}^{\prime \prime }<0$, R can be approximated as a positive constant; the deterministic rate of return C is actually more susceptible to risk perception $\sigma^{2}$. Risk aversion is explained at the core of psychology as “risk taking demands a premium return” [26], while the intertemporal food sales by farmers are based on the pursuit of premium returns. Compared with the relatively fixed risk-averse attitude in food sales, farmers’ perception of the magnitude of risk in intertemporal sales is more helpful in balancing return and risk, which is also in line with the microeconomic assumption of rational individual risk behavior at the same time. As pointed out by Weber et al. [21], individual risk behavior is mainly related to differences in risk perception rather than risk preference. This leads to the following research hypothesis, H_3_:

**H_3_.** 
*Risk preference has less impact on farmers’ food sales decisions than risk perception.*


### 2.2. Analysis of the Moderating Mechanism of Risk Preference

Farmers’ risk decision-making behavior is not only directly affected by risk perception but also by the interaction effect between risk perception and risk preference. The impact of risk perception on farmers’ timing of food sales varies according to risk preference, and combining the analysis of the theoretical framework and farmers’ risk characteristics, the moderating mechanisms of risk preference are mainly based on two paths: theoretical and practical.

First, the theoretical path. From the theoretical mechanisms of related studies, risk preference and risk perception are generally negatively correlated. Vlek and Stallen [27] point out that the higher a person’s risk preference is, the more likely he or she is to underestimate the risk of certain situations, which in turn reduces risk perception. Sitkin and Weingart [19] conclude through behavioral experiments that individuals with a higher risk preference, whose risk perception tends to be lower, and people with a low risk preference may be very sensitive to risk. Specifically, with regard to the timing of food sales, farmers with lower risk preferences will pay extra attention to the risk factors in intertemporal food sales and thus have higher risk perceptions, while farmers with high risk preferences are more likely to recognize and assess positive factors and thus overestimate the probability of obtaining returns from their food sales, and this overestimation of returns directly affects their risk perception of food sales. Meanwhile, based on the analysis of certainty equivalence in the theoretical framework, the juxtaposition of risk preference and risk perception in Equation (7) negatively affects farmers’ choice of intertemporal food sales, and the increase in either risk aversion or the risk perception level will make farmers tend to give up intertemporal food sales, whereas the certainty equivalence of farmers’ expectations is relatively constant when other conditions are unchanged, and there may be a substitution effect between risk preference and risk perception. As R rises, the lower the risk preference, the smaller the decrease in the likelihood of farmers abandoning intertemporal food sales as risk perception increases, i.e., risk preference has a moderating effect on risk perception affecting farmers’ intertemporal food sales.

Second, the practical path. Small farmers in China who rely on food farming are still at the poverty line. Most of them do not have the strength to take risks and are indeed risk averse and have a low risk preference. Farmers with a low risk preference and a high risk perception react more sensitively to possible losses during food sales and are less concerned with less rewarding but relatively secure options. It follows that intertemporal food sales with higher food sales risk may vary significantly with farmers’ risk preference, whereas the difference may not be significant for two-phase food sales with lower food sales risk.

Based on the above analysis of the theoretical and practical paths, research hypotheses H_4_ and H_5_ are formulated as follows:

**H_4_.** 
*Risk preference has a moderating role in the effect of risk perception on farmers’ intertemporal food sales compared to current food sales, i.e., the more risk averse the farmer is, the less the likelihood that the farmer will forego intertemporal food sales decreases as risk perception increases. There is a substitution effect between risk preference and risk perception.*


**H_5_.** 
*Risk preference moderates farmers’ choice of intertemporal food sales significantly compared to current-period food sales, while it does not moderate two-phase food sales significantly.*


Figure 1 illustrates the path of actions of risk preference.

## 3. Data, Modeling, and Variable Selection

### 3.1. Data

This research uses data from the Chinese Family Database (CFD) of Zhejiang University and the China Household Finance Survey (CHFS) conducted by the Survey and Research Center for China Household Finance at the Southwestern University of Finance and Economics, China. Wheat price data are from the National Food and Materials Reserve Administration’s weekly market monitoring reports from 2017 to 2018 and the Bric Agricultural Database from 2015 to 2018.

Due to the availability of data, the Bric Agricultural Database only includes monthly wheat prices in some provinces, and the State Administration of Food and Material Reserves weekly market monitors wheat prices in some regions and years. We chose farmers in the six main wheat-producing provinces of Hebei, Jiangsu, Anhui, Shandong, Henan, and Hubei covered by both databases as the study sample and eliminated the harvested food used exclusively for rations and agricultural production and cases of zero income from food sales, resulting in 1388 samples.

Also, to mitigate the effect of potential outliers, this paper shrinks the continuous variables at the 1% and 99% levels.

### 3.2. Measurement Modeling

In order to examine the effect of risk perception on farmers’ choice of timing of food sales, let *y* denote the timing of food sales chosen by farmers, and the explanatory variable $x_{i}$ varies only with individual *i* and not with mode *j*. It is a multivariate unordered choice problem requiring a control group, and we use a multinomial logit model for empirical estimation. The general form of the model can be expressed as follows:(8)$$P\left(y_{i}=j|x_{i}\right)=\left\{\begin{array}{c} \frac{1}{1+\sum_{k=2}^{J}\exp\left(x_{i}^{\prime }\beta_{k}\right)}\;\;\;\;\;\;\;\;\;\;\;\;\;\;\;\left(j=1\right)\\ \frac{\exp\left(x_{i}^{\prime }\beta_{j}\right)}{1+\sum_{k=2}^{J}\exp\left(x_{i}^{\prime }\beta_{k}\right)}\;\;\;\;\;\;\left(j=2,\ldots ,J\right) \end{array}\right.$$

This model represents the probability that the *i* farmer chooses mode *j*, where *j* is the timing of the food sale and $x_{i}$ is farmer’s risk perception.

### 3.3. Variable Selection and Descriptive Statistics

#### 3.3.1. Dependent Variable

The dependent variable in this paper is the timing of food sales by farmers, the food type is wheat, the choice of intertemporal food sales (in this paper, the raw food processed and sold by farmers is classified as intertemporal food sales) is assigned a value of 1, the choice of two-phase food sales is assigned a value of 2, and the choice of current food sales (in this paper, farmers’ choices for rations + current sales are categorized as current food sales) is assigned a value of 3. The distribution of choices of the sample farmers is reported in Table 1.

Table 1 shows that 89.77% of the sample farmers chose the timing of current sales, indicating that most of the farmers choose to sell all their food after harvesting or leave their rations and then sell them immediately, showing negativity to food sales. The statistical results also reflect that only 1.37% of the sample farmers chose intertemporal sales. Although some farmers will choose to keep rations in the current sales, the part used to reserve for sale is the main body of China’s private food reserves, which is consistent with the conclusion of some scholars that farmers’ food storage in China has declined [28]. In contrast, the proportion of farmers who chose two-phase sales is higher than that of intertemporal sales. The important influence of this difference lies not only in liquidity but also in farmers’ choice of the two, which, to a certain extent, is affected by their risk preference and risk perception. Farmers who choose two-phase sales tend to be risk averse and seek more secure returns, and their risk perception may be relatively high, while farmers who choose intertemporal sales are less risk averse and seek higher returns, and their risk perception may be relatively low. In normal years when food prices are stable, farmers tend to choose intertemporal sales to obtain higher returns. In the case of fluctuating or even inverted prices, farmers may choose two-phase or current sales to avoid risk.

#### 3.3.2. Core Independent Variable

The core independent variable in this paper is farmers’ risk perception. The concept of risk perception belongs to the field of psychology and originated in the theory of “perceived risk”, which was proposed by Bauer at Harvard University for consumer behavior [29]. Subsequently, many scholars have conducted in-depth studies on the basis of this foundation in terms of the nature, dimensions, individual differences, and quantification of risk perception, respectively [30,31,32,33,34,35,36,37]. The more representative ones are by Cunningham, Cox, and Slovic. In particular, Cunningham’s two-factor theory and Cox’s multidimensional theory have been recognized by most scholars and have been widely used in various types of research. The two-factor theory considers risk perception to be composed of the uncertainty of an event and the severity of the adverse consequences. The greatest contribution of this theory is that it builds on Bauer’s theory by quantifying risk perception in both objective and subjective terms [30], and much of the subsequent quantitative research on risk perception has built on this approach. Slovic [17] measured risk perception using a psychometric paradigm based on two-factor theory. Multidimensional theory suggests that risk perception consists of multiple dimensions and that the combination of different dimensional elements realizes a measure of risk perception. Despite the differences in the understanding of risk perception among different scholars, the types of dimensions are basically the same. Table 2 reports the main dimensions of previous research on risk perception.

Combining the analysis of risk dimensions in the classical literature in Table 2, the exploration of farmers’ risk perception in food sales can be categorized into four types: market risk [38,39], liquidity risk [40], transaction cost risk [7,41], and information risk [42]. Based on Equation (9), we refer to Cunningham’s [30] two-factor model and quantify the magnitude of farmers’ risk perception by assigning weight scores to the four types of risk.
(9)$$P=AB=\sum_{i=1}^{n}{\left(a_{ij}\right)}_{m*n}{*\left(b_{ij}\right)}_{n*1}={\left(P_{i}\right)}_{m*1}$$

The two-factor theory considers risk perception P to be composed of A, the uncertainty of an event, and B, the severity of the adverse consequences. The solutions are as follows.

##### The Solution of A

Calculating weights according to the degree of variability of the indicator data can better reflect the uncertainty of risk, and we choose the entropy weight method to objectively quantify the uncertainty of risk. First of all, the risk perception measurement indicators of farmers’ food sales are consistently processed (the indicators have been positively normalized when they are set; see the Appendix A for a description of the indicators), and the matrix $X_{ij}={\left(x_{ij}\right)}_{m*n}$ is as follows:$$X_{ij}=\left[\begin{array}{cccc} x_{11} & x_{12} & \cdots & x_{1n}\\ x_{21} & x_{22} & \cdots & x_{2n}\\ \vdots & \vdots & \ddots & \vdots \\ x_{m1} & x_{m2} & \cdots & x_{mn} \end{array}\right]$$

The matrix $X_{ij}$ is then normalized as follows:$${\tilde{X}}_{ij}={\left({\tilde{x}}_{ij}\right)}_{m*n}=\frac{x_{ij}-\min\{x_{1j},x_{2j},\ldots ,x_{mj}\}}{\max\{x_{1j},x_{2j},\ldots ,x_{mj}\}-\min\{x_{1j},x_{2j},\ldots ,x_{mj}\}}$$

Yield the weight of the *j*th indicator value for the *i*th farmer as follows:$$p_{ij}=\frac{{\tilde{x}}_{ij}}{\sum_{i=1}^{m}{\tilde{x}}_{ij}}$$

Yield the information entropy as follows:$$e_{j}=-\frac{1}{lnm}\sum_{i=1}^{m}p_{ij}\ln(p_{ij})$$

Yield the information entropy redundancy as follows:$$d_{j}=1-e_{j}$$

Calculate the entropy weights for each indicator as follows:$$w_{j}=\frac{d_{j}}{\sum_{j=1}^{n}d_{j}}$$

That is, $A={\left(a_{ij}\right)}_{m*n}=\sum_{j=1}^{n}w_{j}{*({\tilde{x}}_{ij})}_{m*n}$ is obtained. The measurement indexes and weights are shown in Table 3.

##### The Solution of B

Yu L. and Zheng K. [43] pointed out that blindly choosing the objective assignment method may misjudge the importance of the indicators, so this paper chooses the AHP method to subjectively assess the severity of risk through the risk perception dimension. Based on the reality of food sales by farmers, the primary constraint on whether to sell food during the harvest season is liquidity to first meet the financial needs of credit, debt, production, and life. Secondly, most farmers choose to engage in part-time business in order to increase their income, so the transaction cost of intertemporal food sales becomes an important consideration for them. Again, based on the analysis of utility theory, farmers take a negative attitude towards market risk, which is dominated by price fluctuation, and information risk is relatively last. Then, the farmers’ perception of risk severity in descending order is liquidity risk, transaction cost risk, market risk, and information risk. The comparative judgment matrix is constructed according to Saaty’s 1–9 scale method, and the criterion-level judgment matrix is as follows:$$\left[\begin{array}{cccc} 1 & 2 & 3 & 5\\ 1/2 & 1 & 3 & 4\\ 1/3 & 1/3 & 1 & 2\\ 1/5 & 1/4 & 1/2 & 1 \end{array}\right]$$

This matrix CR = 0.0212 < 0.1 meets the consistency test, and the weight vector of the criterion layer is as follows:$${B=(b_{ij})}_{n*1}=\left[\begin{array}{c} 0.4667\\ 0.3146\\ 0.1392\\ 0.0795 \end{array}\right]$$

Combine (1) and (2) for A and B using the definition of risk perception $P=AB=\sum_{i=1}^{m}{\left(a_{ij}\right)}_{m*n}{*(b_{ij})}_{n*1}={\left(P_{i}\right)}_{m*1}$. It can be derived as $P_{i}$, which is the risk perception of the individual farmer.

#### 3.3.3. Moderating Variable

The moderating variable in this paper is the risk preference of farmers. Farmers’ risk preference is measured by the following question: “If you have a sum of money to invest, what kind of investment program are you most willing to choose?” If farmers choose option 1, “high risk and high return”, it means that farmers’ risk preference is risk seeking. Similarly, if farmers choose option 2, “medium risk and medium return”, or option 3, “low risk and low return”, it means that farmers’ risk preference type is risk neutral and risk averse, respectively. The results of the survey show that only 2.02% of the 1388 farmers are risk seeking, while 70.53% are risk averse, i.e., most of the farmers are risk averse to possible risks.

#### 3.3.4. Control Variables

Referring to related studies [37,44], we selected other factors affecting the decision of farmers’ food sales as control variables, including the characteristics of the head of the household (the head of the household’s gender, age, education, physical condition, and whether or not he or she is a village official) and the characteristics of the family (household size, and non-farm income).

#### 3.3.5. Descriptive Statistics

Table 4 shows the definition of each variable and the results of descriptive statistics. The proportion of male heads of the sample households is 66.5%, and the education level of most farmers is elementary school and junior school. The physical condition of the sample is basically general and poor, and 5.5% of the sample households have had the experience of being village officials. The average age of the head of households is about 57 years old, the average household size in the sample households is nearly four persons, and the average household non-farm income is CNY 3730, with the highest household non-farm income being CNY 100,000.

## 4. Analysis of Empirical Results

### 4.1. Benchmark Regression

Table 5 reports the impact of risk perception on farmers’ timing of food sales. Columns (1) and (2) report that relative to current sales, risk perception has negative effects on both intertemporal and two-phase sales at a 1% significance level. In terms of relative risk ratios (RRRs), for each unit increase in risk perception compared with current sales, the odds of farmers choosing intertemporal and two-phase sales decreases to 0.409 and 0.739 times the original ratio, respectively. Columns (3) and (4) report that the risk perception variable remains significant with negative coefficients after adding the control variable (the result is conservative because the realities of credit and storage costs reduce the likelihood that farmers will store for future sales). That is, the higher the risk perception of farmers, the more they are inclined to forego intertemporal and two-phase food sales. Hypothesis H_1_ is verified.

Among the control variables, the gender variable has a significant and positive coefficient on two-phase sales and a positive coefficient on intertemporal sales compared to current sales, which may be due to the fact that male farmers have a higher risk preference compared to female farmers, and even though the perception of risk increases, they still seek to make a profit and thus tend to choose two-phase or intertemporal sales.

Table 6 reports the average marginal effect of the sample (AME). The results show that risk perception has a significant impact on farmer’s food sales. Columns (1) to (3) show, respectively, that for every 0.1 unit increase in risk perception, the probability of farmers choosing intertemporal sales, two-phase sales, and current sales decreases by 11%, decreases by 23%, and increases by 34%, validating the results of benchmark regression.

### 4.2. Endogenous Issues

There is no reverse causality between the dependent and independent variables of the article, but missing variables can have an impact on the estimated results. On the one hand, some farmers have not encountered significant risks in the short term, and the objectivity of risk perception is low; on the other hand, farmers’ food sales decisions may be affected by other unobservable exogenous factors. For this reason, we use the propensity score matching method to mitigate the endogeneity problem between risk perception and the timing of farmers’ food sales.

Firstly, the relevant variables affecting risk perception and farmers’ timing of food sales are treated as covariates to ensure that the negligibility assumption is satisfied. According to the average value of risk perception, the sample is divided into two groups, low risk perception and high risk perception, respectively, which are assigned the values 0 and 1. We use the seemingly uncorrelated model to test the intergroup differences, and the test results in Table 7 show that there are intergroup differences between the different risk perceptions on the timing of food sales by farmers at the 1% significance level, and coefficient comparisons can be made.

The logit model is then used to estimate the propensity score, which is matched using k-nearest neighbor matching and caliper matching methods to estimate the average treatment effect of risk perception on farmers’ timing choices for food sales. Since the traditional propensity matching score method is biased in estimating standard errors, we use the “teffects psmatch” command to derive robust standard errors following Abadie and Imbens [45]. Table 8 reports the estimated average treatment effects and “A-I” robust standard errors. The estimation results show that after controlling for the effects of covariates, a high risk perception significantly and positively influences farmers’ choice of current food sales compared to a low risk perception. Overall, the omitted variables did not seriously interfere with the model estimation results in this paper.

### 4.3. Robustness Check

#### 4.3.1. Replacement of the Dependent Variable

Since the risk perception of farmers is calculated based on the AHP and entropy method weights, in order to test the reliability of the results, this paper replaces the dependent variable with “residents’ happiness”, which takes a value of 1~5, indicating that the degree of unhappiness is increasing in turn, and it uses the same control variables to estimate the impact of risk perception on the “residents’ happiness”. Logically, the higher the risk perception, the lower the resident’ happiness. Since the dependent variable becomes an ordered variable, we estimate it using an ordered logit model. The results, as shown in column (1) in Table 9, show that risk perception positively affects the type of residents’ happiness at the 1% significance level, i.e., the higher the risk perception, the lower the residents’ happiness may be. In summary, the article’s treatment of the dependent variable does not seriously interfere with the robustness of the findings.

#### 4.3.2. Replacement of the Estimation Model

In order to test the robustness of the model, we conduct a robustness test of the baseline regression by changing the model form. Since farmers’ food sales choices move in the opposite direction to their risk perceptions, farmers’ food sales choices can be approximated as an ordered arrangement, so the ordered ologit model is used for replacement. Column (2) in Table 9 shows that risk perception positively affects farmers’ food sales choices at 1% significance level. That is, the higher the risk perception is, the more farmers tend to give up intertemporal sales and choose current sales.

#### 4.3.3. Verification of Marginal Effects

To test the robustness of the marginal effects analysis, we use the mprobit model to verify the impact on marginal effects. Columns (3) and (4) show that both the significance and sign of the variables are consistent with the mlogit model.

Overall, the above tests indicate the robustness of the results of the model analysis used in this paper.

### 4.4. Heterogeneity Analysis

The effect of risk perception on the timing of the food sales of farmers may be significant heterogeneity in different planting scales and the proportion of food sales income to total household income, This paper attempts to reclassify the sample through the characteristics of the planting scale and the percentage of food sales income, through which the heterogeneity of the effect of the timing of the food sales of farmers can be analyzed.

#### 4.4.1. Planting Scale

Farmers with different planting scales have different characteristics in their timing choices for food sales. We divided the sample into three groups, small scale, medium scale, and large scale, based on quartiles and conducted econometric regressions separately, and Table 10 reports the effects of risk perception on the timing choices of the food sales of farmers with different planting scales.

From the regression results, it can be summarized that small-scale farmers’ intertemporal food sales are more affected by farmers’ risk perception, while medium- and large-scale farmers’ two-phase sales are more significantly affected by their risk perception. Small-scale farmers have insignificant returns from intertemporal food sales due to the fragmentation of their plots and tend to give up intertemporal food sales and switch to two-phase or current sales if risk perception increases. Medium- and large-scale farmers, due to their large production scale, insisted on intertemporal sales to pursue returns or chose current sales to avoid risks when risk perception increased.

The above analysis shows that heterogeneity in planting size significantly differentiates farmers’ timing choices for food sales, thus reinforcing the practical basis of the relevant discussion in this paper.

#### 4.4.2. Percentage of Revenue from Food Sales

The percentage of income from food sales to total household income implies the importance of food sales to the farm household in terms of income. There are differences in the characteristics of food sales among farm households with different percentages of income from food sales. We divide the sample into two groups, a low percentage of food sales income and a high percentage of food sales income, based on the median and run separate econometric regressions. Table 11 reports the effect of risk perception on the timing of food sales by farm households with different shares of food sales income.

Meanwhile, the regression results also show that among farmers with a low percentage of income from food sales, intertemporal food sales are more affected by risk perception than two-phase food sales. In the case of farmers with a high share of income from food sales, the impact of risk perception is more significant in the case of two-phase sales. Farmers with a low share of income from food sales do not value the premium income from opportunistic sales, and if they have a high perception of risk after harvesting, they will choose to sell in the current period and give up intertemporal sales and two-phase sales, especially giving up intertemporal sales. When risk perception increases, farmers with a high percentage of income from food sales will insist on intertemporal sales to pursue income or choose current sales to avoid risk, which validates the choice of medium- and large-sized farmers in the heterogeneity of planting size.

To summarize, unlike the attitude of farmers who value food sales with a high percentage of income from food sales, a low percentage of income from food sales is an important factor that induces farmers to be negative toward food sales.

### 4.5. Test of the Moderating Mechanism of Risk Preference

When examining the moderating mechanism of risk preference on risk perception affecting the timing of food sales by farmers, this paper estimated the effects by gradually adding risk preference, the interaction terms of risk preference, and risk perception to the equation. Table 12 reports the effects of risk preference on the timing of farmers’ food sales and the results of the moderating effects.

Columns (1) and (2) normalize risk perception and risk preference with the inclusion of only risk preference and examine the effects of risk perception and risk preference on farmers’ food sales choices. The estimation results show that both risk perception and risk preference significantly and negatively affect farmers’ choices of intertemporal sales and two-phase sales. Hypothesis H_2_ is tested. The standardized coefficients of risk perception in columns (1) and (2) are larger than risk preference, indicating that risk perception has a greater impact on farmers’ food sales choices. Hypothesis H_3_ is tested.

Columns (3) and (4) incorporate the interaction term between risk preference and risk perception. In terms of the timing of food sales, we may categorize intertemporal and two-phase food sales into two groups. Within the group, the estimation in column (3) shows that the coefficients on risk perception and risk preference are both significantly negative, consistent with the estimation in column (1). The coefficient of the interaction term is significantly positive at the 5% confidence level, which not only indicates that risk preference negatively moderates the effect of risk perception on farmers’ intertemporal food sales but also leads to the following inference: there is a relationship between risk preference and risk perception on farmers’ intertemporal food sales, i.e., the substitution effect. Hypothesis H_4_ is verified. Column (4) estimates a non-significant interaction term and a non-significant main effect of risk perception, suggesting that risk preference has no significant moderating role in the effect of risk perception on farmers’ two-phase food sales compared to current food sales.

Between the groups, the test found that the marginal effect of risk perception was −4.8%, which was significant at the 5% level in the intertemporal sales group, where the risk of food sales was higher, and 3.4% in the two-phase sales group, where the risk of food sales was lower, which was not statistically significant. This indicates that in the case of two-phase sales, farmers’ choice of food sales is no longer associated with risk perception. Intuitively, the reverse causal and omitted variables do not differ significantly between the two groups. This suggests that the causal effect of risk perception on farmers’ food sales choices is heterogeneous across risk preference. Further analyzed by simple slope plots, Figure 2 shows that as the *R* value increases, risk preference decreases and the negative selection trend of risk perception on farmers’ inter-phase food sales is significantly weakened, which indicates that there is a substitution effect of farmers’ risk preference on risk perception in intertemporal food sales compared to the current sales. Figure 3 shows that different risk preference groups are parallel to each other, i.e., there is no moderating effect of risk preference in the relationship of risk perception on farmers’ intertemporal food sales. Hypothesis H_5_ is verified.

The analysis of the moderating mechanism shows that risk preference significantly negatively moderates the effect of risk perception on farmers’ intertemporal food sales compared to current food sales, and there is a substitution effect on risk perception, which further confirms that the increase in risk perception is the cause of farmers’ negativity in food sales. There is no moderating effect of risk preference in the effect of risk perception on farmers’ two-phase sales. The causal effect of risk perception on farmers’ timing of food sales is heterogeneous across risk preferences, and the evidence of differences in risk preferences provides a stronger test, showing that a high risk perception contributes to farmers’ negativity regarding food sales.

## 5. Discussion

The effect of risk perception on the timing of food sales of farmers may vary significantly due to different characteristics. The heterogeneity of planting sizes and percentage of food sales revenue allows us to further explore farmers’ sale decisions.

Existing studies generally concluded that large-scale and small-scale farmers tend to choose current sales, while medium-scale farmers tend to choose two-phase sales [44]. Table 10 divides the scale of cultivation into small scale, medium scale, and large scale. The estimation results show that risk perception has a negative effect at a 5% level of significance on small-scale farmers’ choice of intertemporal sales compared to current sales, and the coefficient on two-phase sales is negative but not significant; it is clear that small-scale farmers have a low utility of intertemporal sales due to the small size of cultivation and tend to sell their food directly after harvesting. The results show that medium- and large-scale farmers have similar choices, and the effect of risk perception on their choice of intertemporal food sales is not significant compared to current sales but tends to abandon two-phase sales. The possible reason for this is that some medium- and large-scale farmers, because of the importance of income from food sales, would store food for sale even if there is risk in order to pursue the income from food sales, while some medium- and large-scale farmers are constrained by storage, capital, and other constraints and have difficulty in realizing the sale of food in installments and can only sell it in the current period.

We also want to observe the effect of risk perception on farmers’ grain sale timing decisions at different shares of food sale income in household total income. The estimation results in Table 11 show that risk perception has a negative effect at a 5% level of significance on the choice of intertemporal and two-phase sales by farmers, with a low share of income from food sales compared to current sales, while in the group with a high percentage of income from grain sales, this effect is not so significant, and even the effect on intertemporal sales is not significant. Logically, the higher the percentage of grain sales income in total household income, the greater the importance of grain sales income to the household, so farmers with a low percentage of grain sales income have a stronger willingness to sell directly, while those with a high percentage of grain sales income have to deliberate for the sake of profit.

These findings have two implications. First, compared with large-scale farmers, small-scale farmers tend to sell their food immediately after harvest because the quantity is too little to bring enough benefit. Second, farmers will seriously consider the timing of the sale if food sale income is really important to their family. Conversely, farmers with a low share of income from food sales tend to choose current sales. All in all, farmers’ food sale decisions could be fully explained by expected utility theory, and farmers value risk more than the pursuit of excess returns.

## 6. Conclusions

The focus on food prices under the profit maximization framework in the literature has led many researchers to overlook the importance of sales risk and farmers’ sales decisions. We study Chinese farmers’ timing of food sales and demonstrate that farmers’ negativity in food sales is essentially based on increased risk perception under constraints and the substitution of risk preference. We also find that the effect of risk perception on the timing of food sales has significant heterogeneity in different planting scales and the proportion of food sales income to total household income.

The findings make a number of contributions. First, we break the shackles of price primacy assumptions and reveal the importance of the risk to farmers by utility comparison. The profit maximization theory is generalized while the farmers are more concerned with risk, even though they do not pay much attention to price due to the constraints of risk factors. Risk perception has also led to passive grain sales by peasant households. The dimensions of risk perception are both constraints and pain points in reality, which provide some insights for improving farmers’ situations.

Second, previous studies, including prospect theory, discussed risk perception and risk preference together without directly examining the causal mechanisms of behavioral effects, and our findings reveal the relationship between them in food sales. In terms of farmers’ food sales psychology, unlike the existing research that mainly explains farmers’ economic behavior from the perspective of risk preference, our results validate the point in the debate that risk perception plays a more important role in risk-averse individual decisions, which helps to enrich the literature.

Third, due to risk perception being difficult to measure [46] and data being not trustworthy [7], more studies in recent years involving farmers’ food sale decisions are mainly based on risk preference. While there is some explanatory power, it clearly ignores the important relationship between farmers’ food sales and risk perception. We quantify the risk perception for farmers’ food sales and examine the effects, providing a reference for farmers’ risk research.

Finally, this paper finds that there is a substitution relationship between risk preference and risk perception in the context of farmers’ intertemporal food sales. This finding has important policy implications. While designing policies to support agriculture to reduce farmers’ risk perception, it should also be noted that risk aversion may be increased, resulting in policy failure. Moreover, the substitution effect between risk preference and risk perception is common in other agricultural activities and may raise similar issues.

Our results also suggest the potential importance of experimenting and evaluating policies that focus on farmers’ risk in food sales. Food sales have the most direct and important impact on the ability of farmers to realize the revenue of food production, and most farmers are risk averse towards uncertain returns. While reducing the risk of farmers’ sales, improving the ability to manage risk for farmers is needed. Only by approaching risk and managing it can farmers seize the opportunity to make higher profits and thus break out of the circle of the poor being always poor. In the context of substitution effects, the design of policies to support agriculture to reduce farmers’ risk perceptions should also be cognizant of the potential for elevated risk aversion to cause policy failure.

The study of related issues is of great significance for guaranteeing food supply and promoting common prosperity in the context of rural revitalization. Farmers’ food sale is the first step for food to realize the circulation of the commodity market and increase the income of food farmers. It is also the basis for building a modern food circulation industry. With urbanization, rural aging, hollowing, and the socialization of farm machinery, the burden of ensuring food security will fall on large-scale growers in the near future. Only when the risks are low and the benefits are sufficient will they be willing to grow food well and take responsibility for ensuring food security. If sufficient attention is not paid to the sale of food by farmers, farmers will feel that the food business is risky and low profit, and the incentives for food operation will inevitably suffer.

Since the sample farmers basically have no credit behavior and the storage conditions are almost the same, this paper assumes that the credit and storage conditions of farmers are the same and costless. In the future, we can further consider the storage conditions and the impact of storage losses on food sale behavior. In addition, the interaction mechanism between borrowing costs and farmers’ returns can be examined by the model in this paper. The management of risks to farmers could be replicated in other areas of agriculture.

## Figures and Tables

**Figure 1 foods-13-02243-f001:**
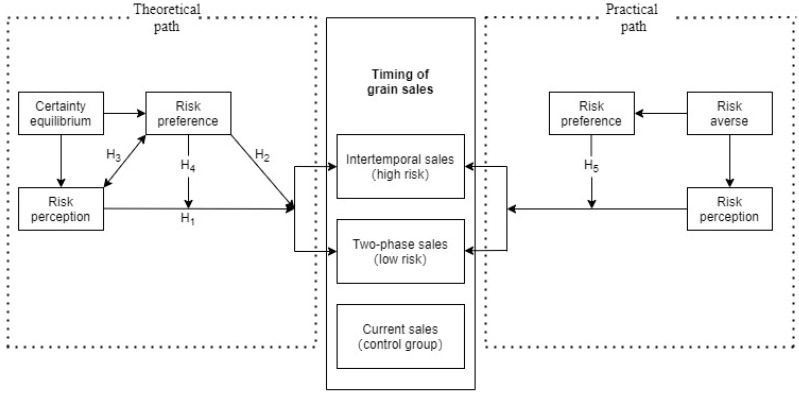
Path diagram of the role of risk preference.

**Figure 2 foods-13-02243-f002:**
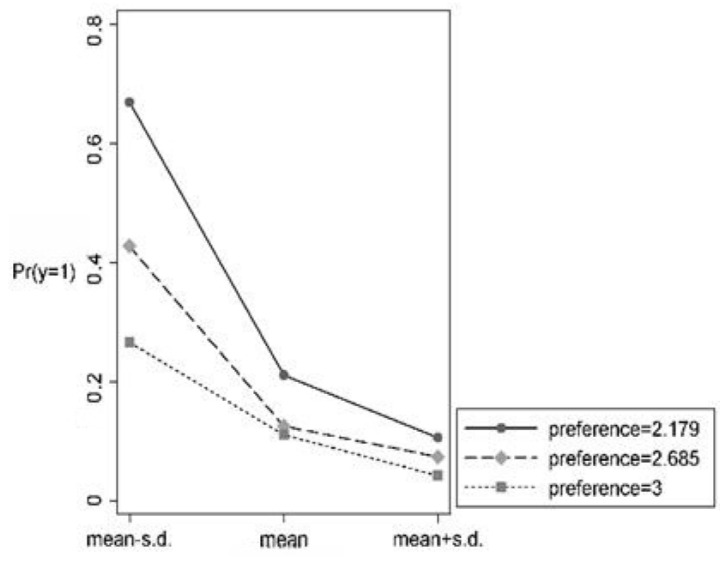
Intertemporal sales simple slope plot.

**Figure 3 foods-13-02243-f003:**
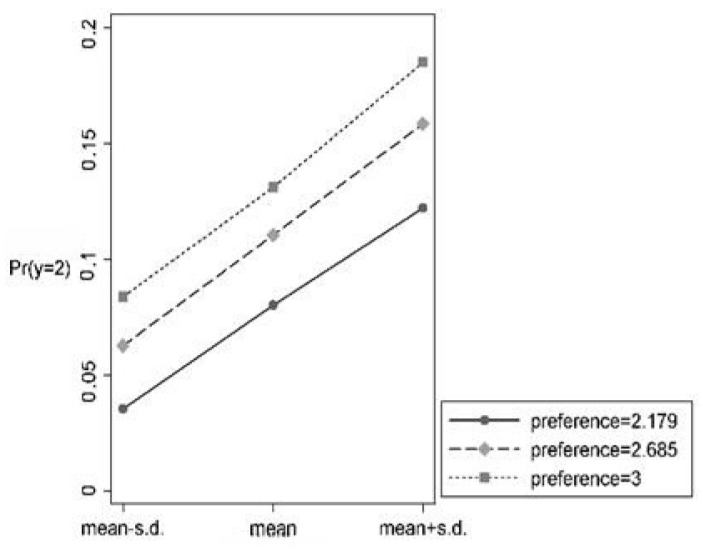
Two-phase sales simple slope plot.

**Table 1 foods-13-02243-t001:** Distribution of the timing of food sales by farmers in six provinces of the main wheat-producing regions.

Timing of Farmers’ Food Sales	Sample Size	Proportions (%)
Intertemporal sales	19	1.37
Two-phase sales	123	8.86
Current sales	1246	89.77

**Table 2 foods-13-02243-t002:** Risk perception research dimensions (it is important to note that since the formulation of risk perception theory is based on consumer behavior, early research dimensions are all about consumers’ risk perceptions).

Research Dimensions	Author [Ref.]
Financial, social, psychological	Cox [31]
Social, money, physical, time, products	Cunningham [30]
Social, intellectual, psychological	Slovic [17]

**Table 3 foods-13-02243-t003:** Indicator weights based on the entropy weighting method.

Risk Perception	Perceptual Factor	Measurement Indicators	Weights (%)
Market risk	Historical experience	Harvest season prices/off-season prices for the last three years	47.22
	Current price	Price ratio for the same period	42.15
	Price expectations	Degree of market concern	10.63
	Fluid funds	Amount of cash	54.12
Liquidity	Current ratio	Consumption expenditure/gross income	28.16
risk	Percentage of revenue from food sales	Revenue from food sales/total revenue	17.72
	Sales channel	Number of sales channels	65.24
Transaction	Sales objects	Whether to sell only to rural food merchants	31.36
cost risk	Costs of transportation, hired labor, etc.	Number of relatives and neighbors helping	3.40
Information risk	External information	Whether to buy online	50.21
	Local information	Availability of agricultural guidance	49.79

**Table 4 foods-13-02243-t004:** Variable definitions and descriptive statistics.

Variable	Definitions	Mean	S.D.
Timing of food sales	Intertemporal, 1.37%; two phase, 8.86%; current, 89.77%	—	—
Risk preference	Risk pursuing, 2.02%; risk neutural, 27.45%; risk averse, 70.53%	—	—
Gender	Male, 66.5%; female, 33.5%	—	—
Education	Illiterate, 21.61%; primary, 34.87%; junior, 33.07%; middle, 15.63%; college, 0.6%	—	—
Physical condition	Very good, 9.08%; good, 16.35%; general, 35.73%; poor, 31.56%; very poor, 14.70%	—	—
Village officials	Yes, 5.5%; No, 94.5%	—	—
Risk perception (to avoid making the relative risk ratios too small for labeling, this paper multiplies the risk perception by 10)	Farmers’ perceived level of risk in food sales	4.38	1.24
Age	Age of head of household	57.34	11.00
Household size	Number of family members	3.52	1.70
Non-farm income	Income not from agriculture, unit: CNY ten thousand	0.37	1.09

**Table 5 foods-13-02243-t005:** Impact of risk perception on farmers’ timing of food sales.

	(1)		(2)		(3)		(4)	
	Coef.	RRR	Coef.	RRR	Coef.	RRR	Coef.	RRR
Risk perception	−0.893 ***	0.409	−0.303 ***	0.739	−0.865 **	0.421	−0.304 ***	0.738
	(0.343)		(0.111)		(0.404)		(0.106)	
Gender					0.357	1.430	0.409 *	1.505
					(0.595)		(0.237)	
Age					−0.002	0.998	0.002	1.002
					(0.029)		(0.011)	
Education					0.398	1.488	−0.155	0.857
					(0.249)		(0.115)	
Physical condition					0.216	1.241	0.019	1.019
					(0.136)		(0.050)	
Village officials					−0.241	0.786	−0.278	0.758
					(1.062)		(0.494)	
Household size					0.123	1.131	−0.104	0.901
					(0.126)		(0.065)	
Non-farm income					−0.495	0.610	0.136	1.145
					(0.433)		(0.085)	
Region					Control		Control	
Constant term	−0.661	0.517	−1.037	0.355	−3.571	0.028	−1.073	0.342
	(1.248)		(0.459)		(2.354)		(1.057)	
Observations	1388				1388			
Wald chi2	13.55				986.04			
Prob > chi2	0.0011				0.0000			
Pseudo R^2^	0.0228				0.0659			

Notes: Robust standard errors in parentheses; ***, **, and * indicate significance at 1%, 5%, and 10% levels, respectively.

**Table 6 foods-13-02243-t006:** The average marginal effect of risk perception on farmers’ timing of food sales.

	(1)	(2)	(3)
Risk perception	−0.011 **	−0.023 ***	0.034 ***
	(0.006)	(0.008)	(0.010)
Control variable	Control	Control	Control
Observations	1388	1388	1388

Note: Robust standard errors in parentheses; ***, ** indicate significance at 1%, 5% levels, respectively.

**Table 7 foods-13-02243-t007:** Results of the test for between-group differences based on the seemingly uncorrelated model.

	Intertemporal Sales		Two-Phase Sales	
	Low Perception	High Perception	Low Perception	High Perception
Risk perception	−1.155 *** (0.341)	49.193 *** (3.612)	−0.693 *** (0.203)	0.346 (0.340)
Control variable	Control	Control	Control	Control
Region	Control	Control	Control	Control
Observations	1388		1388	
Intergroup regression coefficients	192.55		6.89	
difference-in-difference test	0.0000		0.0087	

Notes: *** indicate significance at 1% level, respectively.

**Table 8 foods-13-02243-t008:** Estimation based on the propensity score matching method.

	Parameters	Average Treatment Effect	“A-I” Robust Standard Errors
k-nearest neighbor matching	k = 1	0.073 **	0.029
	k = 4	0.084 ***	0.023
Caliper matching	Caliper = 0.1	0.084 ***	0.021

Notes: ***, ** indicate significance at 1%, 5%levels, respectively.

**Table 9 foods-13-02243-t009:** Robustness check.

	(1)	(2)	(3)	(4)
	Coef.	Coef.	AME	AME
Risk perception	0.111 ***	0.367 ***	−0.009 **	−0.022 ***
	(0.043)	(0.111)	(0.005)	(0.007)
Control variable	Control	Control	Control	Control
Region	Control	Control	Control	Control
Observations	1388	1388	1388	1388
Wald chi2	154.72	46.19	——	——
Prob > chi2	0.0000	0.0000	——	——
Pseudo R2	0.0482	0.0446	——	——

Note: Robust standard errors in parentheses; ***, ** indicate significance at 1%, 5% levels, respectively.

**Table 10 foods-13-02243-t010:** Analysis of heterogeneity across planting sizes.

	Small-Scale Farmers		Medium-Scale Farmers		Large-Scale Farmers	
	Intertemporal	Two Phase	Intertemporal	Two Phase	Intertemporal	Two Phase
Risk perception	−0.820 **	−0.106	−0.992	−0.461 **	−0.703	−0.363 **
	(0.411)	(0.160)	(1.093)	(0.208)	(0.677)	(0.174)
Control variable	Control		Control		Control	
Region	Control		Control		Control	
Observations	531		464		393	
Wald chi2	1740.49		29.37		2850.59	
Prob > chi2	0.0000		0.2945		0.0000	
Pseudo R2	0.1093		0.1018		0.1351	

Note: Robust standard errors in parentheses; ** indicate significance at 5% levels, respectively.

**Table 11 foods-13-02243-t011:** Analysis of heterogeneity in the share of income from different food sales.

	Low Percentage of Income from Food Sales		High Percentage of Income from Food Sales	
	Intertemporal	Two Phase	Intertemporal	Two Phase
Risk perception	−0.742 **	−0.318 **	−0.754	−0.299 *
	(0.304)	(0.147)	(0.672)	(0.153)
Control variable	Control		Control	
Region	Control		Control	
Observations	645		743	
Wald chi2	2105.14		1721.52	
Prob > chi2	0.0000		0.0000	
Pseudo R2	0.0863		0.1207	

Note: Robust standard errors in parentheses; **, and * indicate significance at 5%, and 10% levels, respectively.

**Table 12 foods-13-02243-t012:** Moderating effect of risk preference.

	(1)		(2)		(3)		(4)	
	Coef.	RRR	Coef.	RRR	Coef.	RRR	Coef.	RRR
Risk perception	−6.776 **	0.001	−2.071 ***	0.126	−3.820 ***	0.022	0.374	1.454
	(3.209)		(0.737)		(1.434)		(0.438)	
Risk preference	−1.476 *	0.228	−0.832 **	0.435	−4.872 ***	0.008	0.677	1.968
	(0.837)		(0.378)		(1.865)		(0.731)	
Interaction term	——	——	——	——	1.172 **	3.228	−0.269	0.764
	——	——	——	——	(0.531)		(0.171)	
Controls	Control		Control		Control		Control	
Region	Control		Control		Control		Control	
Observations	1388				1388			
Wald chi2	740.40				241.22			
Prob > chi2	0.0000				0.0000			

Notes: Robust standard errors in parentheses; ***, **, and * indicate significance at 1%, 5%, and 10% levels, respectively.

## Data Availability

The original contributions presented in the study are included in the article, further inquiries can be directed to the corresponding author.

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
