# Peer review of "Risk Perception, Risk Preference, and Timing of Food Sales: New Insights into Farmers’ Negativity in China"

_foods, 2024, doi:10.3390/foods13142243_

Round 1

Reviewer 1 Report

Comments and Suggestions for Authors

Dear Authors,

The manuscript entitled “Risk Perception, Risk Preference and Timing of Food Sales: New Insights into Farmers’ Indifference in China” deals with an interesting and current topic. It has, however, some major issues to be handled.

First, please, revise the sentence in lines 30-31; how could a battle be won against poverty when there is a huge number of people living in poverty? It is not clear what you mean by “whether from academia of practice” in the given context. A serious issue is the incorrect use of the term “indifferent”. It means neutral, i.e., it does not matter for the person which option he/she chooses. But you use it in the meaning of preference for selling earlier/immediately. So please, change it everywhere in the manuscript (including the title). A reference is needed in lines 47-48. You should not mention your results in the Introduction section; moreover, lines 97-112 belong to the Conclusions section. It is not clear what you mean by “risk perception from the side” in line 92. The term “mechanistic analysis” is not appropriate in the given context. You mention two-phase food sales in line 184 for the first time but explain its meaning online in a footnote related to line 280 after mentioning it several times. Please, revise Figure 1, since it is not clear (e.g., what does “No moderate” mean?). Please, insert a figure containing the whole model with the remaining hypotheses. The size of the study population is needed to be presented. References are needed in lines 304 and 308. Your chosen risk categories have almost nothing to do with the dimensions listed in Table 2, so you cannot state “Based on the analysis […] in Table 2”. It is not clear in advance what you mean by “The Solution of A” and “The Solution of B”, since neither A nor B has been defined previously. It is not clear what you mean by “early research dimensions tended to be biased towards consumers’ risk perceptions”; of course, they tended to be, since it is all about consumers’ risk perceptions. Please, specify the AHP method. When measuring farmers’ risk preference, you could not receive reliable answers, since the answer depends on the sum of money they invest (as you arrived at this conclusion later in your own analysis). It is not clear how the average household non-farm income could be 0.373 million yuan (=373,000 yuan), when the highest household non-farm income is only 100,000 yuan. Table 4 contains a lot of incorrect values; several variables are nominal and ordinal in nature; therefore means and standard deviations cannot be calculated. Instead, provide frequencies. The non-farm income variable does not affect farmers’ choice positively, since it is not significant (see lines 414-418). Why are the columns in Table 6 named (5), (6), and (7) instead of (1), (2), and (3)? It is not clear why you always include “Control variable” with “Control” values in Tables 6, 7, 8, 9, 10, 11, and 12. Similarly, “Region” has not even been mentioned before Table 7, where it appears as “Control”, and then in the remaining tables as well. Please, revise the sentence in lines 428-432 and 541-552; it is not totally clear what you mean there. Lines 499-501 belong to the literature review, not the results. Explanations of results (e.g., around lines 506-516 and 541-552) belong to the Discussion section (which is missing now). Please, take a stand in case of H2 and H3 in line 582 (it is not enough to state that they have been tested). Please, add names to the axis and lines on Figures 2 and 3. The Appendix contains several questionable (or unclear) statements. It is not clear, e.g., what you mean by “Sell relatively lagging, so choose June, July and August”. It is also not clear how could time be from June 2018 to June 2017 (mentioned twice). How could the price risk be the same a year later as that of the purchase price? There are several variables that are not included in the models; so it is not clear why they are discussed in the Appendix (see the level of market interest, 1/amount of cash, consumption expenditure/gross income, 1/number of sales channel, whether to sell only to rural food merchants, number of relatives and neighbors helping, availability of agricultural guidance, whether to buy online). Moreover, the question for the variable 1/number of sales channels (“What are the marketing channels…”) is not related to the variable, since it is not about the number of the channels. It is also not clear how the number of online shopping is related to the sufficient level of access to outside information and the risk of information loss. The formatting of in-text citations does not meet the journal’s requirements.

Comments on the Quality of English Language

There are some spelling/grammar mistakes in the manuscript, see , e.g., lines 14, 174, 207, 304, footnote 1 (p.7), line 399, Table 13 (several unnecessary capital letters, commas, spaces, etc.).

Reviewer 2 Report

Comments and Suggestions for Authors

Dear authors,

Your paper is fine, but what is most important is missing. Please highlight the discussions, implications, comparisons with previous papers. Try to focus on the managerial contributions, explain the utility of your study and how can it be replicated in other areas.

Reviewer 3 Report

Comments and Suggestions for Authors

Dear Dr. Tian:

I would like to ask if you have interviewed any farmers or have any idea why they sell in this case their wheat as soon as possible.  Please rewrite your abstract to reflect reality.

L 98 is this your “hedge”?  If this can be a reason?

L 50, L130 From my experience your premise that “farmers are averse to risk” is a fundamental error.

L 86 Subsistence farmers in India were able to break the stranglehold “brokers” had on their fruit crops under a similar situation when they had access to cell phones and could bypass the system of middlemen.  New technology is undermining your main premises.

L 239 I think that a vast majority of these wheat farmers could tell you to a fine degree what the price of wheat on the “cash market” was for all of last year.  

L 256  Does it matter that your data is 8 to 9 years out of date?

L 288—Please consider another “causation factor” other than risk avoidance that caused nearly 90% of these farmers to see immediately after harvest.

Table 4 and others.  Please do not report mean with 4 significant figures.  You have variation, SD in the tenths and units place.

L 537—550 —This is good.   

L 628—Did you take into consideration a) Did the farmer own the land or were they renting and had to make payments to the landlord?  b) what was the level of indebtedness?  Did the farmer owe a bank or other income from this year’s harvest?

L 638 Seems to be a hedge.  What is the likelihood that any of the persons involved will benefit from your analysis?  What is the likelihood that policymakers will change their long-held positions based on your analysis?

Round 2

Reviewer 3 Report

Comments and Suggestions for Authors

Dr. Tian:

First line in abstract:  In your response to my suggestions, you basically said there were large, medium and small (submarginal) groups of farmers.  The timing of selling was different among the three groups.  I think your first line is an over generalization based on your data.  See your footnote on page 4 and L 573

You have added “negative and negativeness” throughout the manuscript—please clarify what you intend by this early on in the manuscript

L 80 Good addition

L 277 Capitalize University

L 581 remove the hyphen

> L 591 much improved

> 632 Some empathetic thoughts
